# Chemical and Biological Evaluation of Novel 1*H*-Chromeno[3,2-*c*]pyridine Derivatives as MAO Inhibitors Endowed with Potential Anticancer Activity

**DOI:** 10.3390/ijms24097724

**Published:** 2023-04-23

**Authors:** Larisa N. Kulikova, Rosa Purgatorio, Andrey A. Beloglazkin, Viktor A. Tafeenko, Raesi Gh. Reza, Daria D. Levickaya, Sabina Sblano, Angelina Boccarelli, Modesto de Candia, Marco Catto, Leonid G. Voskressensky, Cosimo D. Altomare

**Affiliations:** 1Organic Chemistry Department, Peoples’ Friendship University of Russia (RUDN University), 6 Miklukho-Maklaya St., 117198 Moscow, Russia; l-n.kulikova@yandex.ru (L.N.K.); drlevi@rambler.ru (D.D.L.); lvoskressensky@sci.pfu.edu.ru (L.G.V.); 2Department of Pharmacy-Pharmaceutical Sciences, University of Bari Aldo Moro, Via E. Orabona 4, 70125 Bari, Italy; rosa.purgatorio@uniba.it (R.P.); sabina.sblano@uniba.it (S.S.); modesto.decandia@uniba.it (M.d.C.); marco.catto@uniba.it (M.C.); 3A.V. Topchiev Institute of Petrochemical Synthesis, Russian Academy of Sciences, 29 Leninskiy Prosp., 119991 Moscow, Russia; 4Department of Chemistry, Lomonosov Moscow State University, Leninskie Gory 1-3, 119234 Moscow, Russia; 5Department of Precision and Regenerative Medicine and Ionian Area, School of Medicine, University of Bari Aldo Moro, Piazza Giulio Cesare 11, 70124 Bari, Italy; angelina.boccarelli@uniba.it

**Keywords:** 1*H*-chromeno[3,2-*c*]pyridine, monoamine oxidases A and B, MAO inhibitors, multitarget directed ligands, neuroprotection, antitumor activity

## Abstract

About twenty molecules sharing 1*H*-chromeno[3,2-*c*]pyridine as the scaffold and differing in the degree of saturation of the pyridine ring, oxidation at C10, 1-phenylethynyl at C1 and 1*H*-indol-3-yl fragments at C10, as well as a few small substituents at C6 and C8, were synthesized starting from 1,2,3,4-tetrahydro-2-methylchromeno[3,2-*c*]pyridin-10-ones (1,2,3,4-THCP-10-ones, **1**) or 2,3-dihydro-2-methyl-1*H*-chromeno[3,2-*c*]pyridines (2,3-DHPCs, **2**). The newly synthesized compounds were tested as inhibitors of the human isoforms of monoamine oxidase (MAO A and B) and cholinesterase (AChE and BChE), and the following main SARs were inferred: (i) The 2,3-DHCP derivatives **2** inhibit MAO A (IC_50_ about 1 μM) preferentially; (ii) the 1,2,3,4-THCP-10-one **3a**, bearing the phenylethynyl fragment at C1, returned as a potent MAO B inhibitor (IC_50_ 0.51 μM) and moderate inhibitor of both ChEs (IC_50_s 7–8 μM); (iii) the 1*H*-indol-3-yl fragment at C10 slightly increases the MAO B inhibition potency, with the analog **6c** achieving MAO B IC_50_ of 3.51 μM. The MAO B inhibitor **3a** deserves further pharmacological studies as a remedy in the symptomatic treatment of Parkinson’s disease and neuroprotectant for Alzheimer’s disease. Besides the established neuroprotective effects of MAO inhibitors, the role of MAOs in tumor insurgence and progression has been recently reported. Herein, antiproliferative assays with breast (MCF-7), colon (HCT116) and cisplatin-resistant ovarian (SK-OV-3) tumor cells revealed that the 10-indolyl-bearing 2,3,4,10-THCP analog **6c** exerts anti-tumor activity with IC_50_s in the range 4.83–11.3 μM.

## 1. Introduction

Chromone (i.e., 4*H*-chromen-4-one or 4*H*-1-benzopyran-4-one), a widely recurring oxaheterocyclic moiety of plant nature (e.g., flavonoids and xanthones), has gained importance as a ‘privileged’ scaffold in medicinal chemistry, due to its drug-likeness and diverse ascertained pharmacological activities, such as neuroprotectant, anticancer, antimicrobial, antiviral, anti-inflammatory and antioxidant activities [1]. Simple and fused chromones’ derivatives showed potential in the design of multitarget-directed ligands (MTDLs) for treating neurodegenerative disorders, i.e., Alzheimer’s (AD) or Parkinson’s disease (PD), as inhibitors of acetylcholinesterase (AChE) and butyrylcholinesterase (BChE), monoamine oxidases A and B (MAO A and B) and amyloid beta (Aβ) plaque formation and aggregation [2,3]. Moreover, the exploitation of chromone-containing heterocycles as privileged scaffolds in anticancer drug discovery is supported by the evidence that naturally occurring and synthetic chromones are endowed with antitumor activity, through diverse mechanisms (cytotoxicity, antimetastasis, antiangiogenesis, chemoprevention, etc.) [4].

Among the chromone-annulated derivatives, we recently focused on the medicinal chemistry of piperidine-fused chromone derivatives and investigated in vitro their potential as anti-AD agents [5]. In our previous work, a number of novel 2-alkyl derivatives of 1,2,3,4-tetrahydrochromeno[3,2-*c*]pyridin-10-one (**1**, Figure 1), bearing different substituents at C6 and C8, were synthesized and characterized as MTDLs against MAOs A and B, AChE and BChE, which are targets of drugs alleviating the symptoms of neurodegenerative dementias, including AD. The synthesized compounds achieved in several cases’ inhibitory potencies in the low-to-submicromolar range, with isoform selectivity toward MAO B and AChE. In particular, compound **1** (Figure 1), bearing Bn and 6-OEt as R and R^1^ substituents, respectively, proved to be MAO B-selective (IC_50_ 0.89 μM), whereas the same scaffold **1**, bearing *i*Pr as R and 8-Br as R^1^, achieved single-digit micromolar potency against either MAO B (IC_50_ 2.23 μM) and AChE (IC_50_ 3.22 μM).

Herein, relying on this evidence, we wanted to further explore the reactivity of 1,2,3,4-tetrahydro-2-methylchromeno[3,2-*c*]pyridin-10-ones (1,2,3,4-THCP-10-ones **1**) and 2,3-dihydro-2-methyl-1*H*-chromeno[3,2-*c*]pyridines (2,3-DHCP **2**), generating novel chromeno[3,2-*c*]pyridine derivatives (**3**, **6**–**8**), which may ultimately expand the toolbox of MAOs and ChEs inhibitors with potential utility as neuroprotectants in AD and/or PD. A selection of compounds, endowed with good inhibitory potency towards MAO B, was also assayed for the tumor growth inhibitory activity in three tumor cell lines, namely breast (MCF-7), colon (HCT116) and cisplatin-resistant ovarian (SK-OV-3) tumor cells.

## 2. Results and Discussion

### 2.1. Chemistry

1,2,3,4-tetrahydrochromeno[3,2-*c*]pyridin-10-one (**1**) and 2,3-dihydrochromeno[3,2-c]pyridines (**2**), whose scaffold will be henceforth referred as THCP and DHCP, respectively, were synthesized through a three-stage synthesis. The synthesis of the THCP-10-one derivatives **1** was described earlier [5,6,7]. The synthesis of the DHCPs **2** was carried out through condensation of morpholine with *N*-methylpiperidone followed by cyclization of the resulting enamine with substituted salicylic aldehydes and boiling of the intermediate hexahydrochromeno[3,2-*c*]pyridine in *o*-xylene (Figure 1).

In this study, the introduction of X-substituted ethinyl fragments onto the tetrahydropyridine cycle expanded the synthetic and biological capabilities of the chromenopyridines. 1,2-Dihydrochromeno[3,2-*c*]pyridin-10-one (DHCP-10-one) derivatives **3a**–**d** were obtained by a cross-coupling reaction of **1a**–**d** with terminal alkynes in the presence of CuI and diisopropylazadicarboxylate (DIAD) (Figure 2).

The reaction was conducted in dry THF, according to the previously reported synthetic procedure [8]. Most likely, in the first stage, the dihydropyridine cycle is oxidized under the action of DIAD to the pyridinium salt B, which is then alkynylated with copper acetylenide formed from the corresponding alkyne and CuI (Figure 3).

The molecular structure of **3a** was confirmed by X-ray diffraction analysis (Figure 1).

2,3-Dihydrochromeno[3,2-*c*]pyridines (**2a** and **2b**) were functionalized with various nucleophiles under acid catalysis conditions. *N*-methyl pyrrole, nitromethane and indole or 5-substituted indoles were used as C-H nucleophiles. The reactions proceeded in CF_3_CH_2_OH under microwave-activation conditions. All nucleophilic addition products were isolated by column chromatography (Figure 4).

We supposed that the reaction proceeds through the acid-catalyzed transformation of the DHCP cycle to benzopyryl salts. The latter readily takes part in the reactions of nucleophilic addition. The probable mechanism of these transformations is shown in Figure 5.

Recently, some of us proved that 12*H*-chromeno[2,3-*c*]isoquinoline bearing the 1*H*-indolyl group at C12 has antiproliferative activity against diverse human tumor cells (i.e., MCF-7, HCT116, A2780, SK-OV-3) with IC_50_s in the low micromolar range [9]. Due to a certain degree of molecular similarity with that compound of the indole-bearing THCP derivatives **6a**–**c** synthesized in this study, we tried to optimize a simpler procedure for their synthesis. Earlier the synthesis of similar compounds had been accomplished through multicomponent reactions of salicylic aldehydes with C-H acids and various nucleophiles [10,11,12]. The key factors for the reactions’ success were the catalytic system, temperature, and the type of process activation.

The interaction of salicylic aldehyde, 4-methylpiperidine and 1*H*-indole was performed in EtOH under heating in the presence of 10 mol% l-proline. However, instead of the expected product **6**, hemiacetal **7** was obtained as a white crystalline substance (Figure 6). We tried to optimize the reaction conditions by varying solvents, temperatures and catalysts (Table 1). The expected product **6** was obtained in just 12% yield only in CF_3_CH_2_OH as the solvent.

While we failed in finding the conditions for the three-component synthesis of **6a**, it is worth highlighting that the resulting cyclic hemiacetal **7a** has not been described anywhere before and it was of interest as a new derivative of the chromene series. Keeping this in mind, we obtained a series of **7a**–**d** hemiacetals by reactions of salicylic aldehyde and its substituted derivatives with *N*-methylpyrrolidone in EtOH in the presence of l-proline as the catalyst. The products were crystallized and isolated by filtration (Figure 7).

The structure of the cyclic hemiacetal **7c** was confirmed by X-ray diffraction data (Figure 2), which allowed us to establish that two hydroxyl groups of the central cycle have relative trans-configuration, whereas the azadecalin system of fused two non-aromatic six-membered cycles has a cis-configuration (Figure 3). In this way, the compound having 4aR*, 10S*, 10aR* configuration of asymmetric centers is formed in the reaction diastereoselectively.

It should be noted that according to the X-ray data, compound **7c** is formed as a racemate (the spatial symmetry group of the formed crystal P21/C, i.e., centrosymmetric). This means the co-crystallization of both enantiomers in one crystal.

Probably, the first stage of the mechanism of hemiacetals **7a**–**f** formation involves the reaction of a salicylic aldehyde with l-proline yielding the intermediate A. The latter is nucleophilically attacked by the enol form of piperidone to form intermediate B, the conformation of which is fixed by the formation of HB between the protonated carbonyl group of piperidone and the carboxylate group of proline. Thus, free rotation around the bond between carbon in α-position of the benzene ring and piperidine fragment (marked on the scheme) is blocked yielding cis-decalin system of the cyclic hemiacetals **7** (Figure 8).

To confirm the role of l-proline in the stereoselective formation of **7**, we performed reactions of salicylic aldehydes with 4-methylpiperidone without its addition. As a result, a series of compounds **8a**–**e** was obtained with almost the same yields as for reactions in the presence of proline (Figure 9).

However, the stereochemistry of compound **8** differs from the stereochemistry of compound **7** (Figure 3). The two hydroxyl groups of the central cycle have the same configuration as in the case of compound **8a**, but the fusion of six-membered cycles of the azadecalin system has trans-configuration (Figure 4). Thus, the reaction diastereoselectively yields compounds with a relative 4aR*,10S*,10aS* configuration of asymmetric centers.

It is worth mentioning that the ^1^H NMR spectra of the isomeric **7** and **8** are very close having similar chemical shifts of protons signals and spin–spin coupling constants (the difference in chemical shifts of the corresponding protons is no more than 0.01 ppm). However, these compounds have different melting points (values in Table 2).

Such diastereoselectivity in the synthesis of hemiacetals **8** could be explained by the formation of the more thermodynamically preferable trans-decalin system due to the possibility of the attack of phenolic hydroxyl from both possible sides in intermediates B and C (Figure 10).

### 2.2. Biological Evaluation

#### 2.2.1. Inhibition of Monoamine Oxidases and Cholinesterases

Diverse 1*H*-chromeno[3,2-*c*]pyridine derivatives synthesized herein, with different saturation degree of the pyridine ring and substituents (R, R^1^, R^2^ and X), were tested as inhibitors of the human isoforms of MAO (A and B) and ChE (AChE and BChE). To extend the knowledge of the structure–activity relationships (SARs) of this class of annulated oxaza-heterocyclic derivatives, the activities of the previously reported 1,2,3,4-tetrahydrochromeno[3,2-*c*]pyridin-10-one (1,2,3,4-THCP-10-one) **1a** were evaluated and compared with those of 2,3-dihydro-1*H*-chromeno[3,2-*c*]pyridines (2,3-DHCP) **2a**–**b**, 2-phenylethynyl-substituted 1,2-dihydrochromeno[3,2-*c*]pyridin-10-ones (1,2-DHCP-10-ones) **3a**–**d** and 10-indolyl-substituted 2,3,4,10-tetrahydro-1*H*-chromeno[3,2-*c*]pyridine (2,3,4,10-THCP) **6a**–**c**. The MAO B-selective inhibitor pargyline and the AChE-selective inhibitor galantamine were used as positive controls. All of the compounds were tested against each enzyme at a single 10 μM concentration, and for those showing more than 60% inhibition at that concentration, IC_50_s were determined in at least three independent experiments (Table 3).

Except for the 1-(2-phenylethynyl) derivative of 1,2-DHCP **3a**, which achieved single-digit micromolar IC_50_ values against both ChEs, all the newly synthesized chromeno[3,2-*c*]pyridine derivatives resulted scarcely active as ChEs inhibitors. 2,3-DHCPs **2a** and **2b** returned good MAO A inhibition data (IC_50_s 1.18 and 0.703 μM, respectively) and about tenfold selectivity over MAO B, whereas the MAO inhibitory activity of the 1,2-DHCP-10-ones **3a**–**c** was affected by the substituents at N2 and C1. Within the limits of the explored property space around the 1,2-DHCP-10-one scaffold, the pattern 2-methyl/1-phenylethynyl (**3a** and **3b**) solely warranted submicromolar MAO B inhibition and more than twentyfold MAO B/A selectivity, while bulkier substituents at N2 caused a drop of the inhibitory potency. These results are congruent with those obtained from a very recent exploration of the 1,2,3,4-tetrahydrobenzo[*b*][1,6]naphthyridine scaffold of novel MAO inhibitors [13]. Indeed, the 1-(2-(4-fluorophenyl)ethynyl)-2-methyl analog proved to be in vitro a potent MAO B inhibitor with IC_50_ of 1.35 μM. Comparing the MAO B inhibition data of **3a**–**b** with those of **3c**–**d**, and **1a** as well, it appears that (i) the phenylethynyl group at C1 may be accommodated into the MAO B binding site better than into that of MAO A, (ii) the presence of the OMe or OEt substituents at C6 does not affect the MAO B inhibition, whereas (iii) alkyls (Et, *i*Pr) bulkier than the Me group on N2 cause at least a twentyfold decrease in MAO B inhibition potency.

Moreover, the 2,3,4,10-THCP compounds **6a**–**c**, bearing a 1*H*-indol-3-yl group at C10, achieved a fair activity as MAO B-selective inhibitors (IC_50_s ranging between 3.51 and 7.30 μM), with a slightly favorable lipophilic effect of the substituent R^2^ at C5’ position of the 1*H*-indole moiety (Br > OMe > H).

We sampled a few hemiacetals, three with cis-ring fusion (**7a**,**c**,**e**) and three with trans-ring fusion (**8a**,**c**,**e**), and assayed them for MAOs and ChEs inhibitory activities (Table 4). Regardless, the ring fusion stereochemistry (4aR,10S,10aR for compounds **7** and 4aR,10S,10aS for compounds **8**), position and lipophilicity of the substituents, all the cyclic hemiacetals resulted poor inhibitors of MAO B with IC_50_s > 10 μM and in most cases even poorer inhibitors of AChE. The inhibitory activity of all the tested hemiacetals toward MAO A and BChE at 10 μM concentration was found to be very weak or null.

The investigation of the inhibition kinetics of **3b** and **6c**, taken as representative of the two subsets of MAO B-selective inhibitors, resulted in Michaelis-Menten curves’ fitting for competitive MAO B inhibition (Figure 5), with inhibition constant (*K*_i_) values equal to 1.41 ± 0.21 μM and 6.47 ± 0.22 μM, respectively.

#### 2.2.2. Antiproliferative Activity on Tumor Cell Lines

Some years ago, some of us investigated the antiproliferative activity of 12*H*-chromeno[2,3-*c*]isoquinoline derivatives [9], among which the analog bearing 1*H*-indol-3-yl moiety at C12 achieved a 50% inhibition of cell growth in some tumor cell lines, including the cisplatin-resistant ovarian carcinoma one, in the low micromolar range. Considering the molecular similarity of that compound with 10-(1*H*-indol-3-yl)-bearing 2,3,4,10-THCP analogs (**6**), we assayed compounds **6a**–**c**, along with **2a**–**b** and **3b**–**c**, in three human tumor cell lines, i.e., breast (MCF-7), colon (HCT116) and ovarian resistant (SK-OV-3) tumor cells. Cisplatin (CDDP) and doxorubicin (DXR) were used as positive controls. Besides the antitumor activity, we were also interested in challenging recent studies supporting the role of monoamine oxidases in tumor proliferation [14]. Such evidence would give new chances to MAO inhibitors for being repositioned as coadjutants in the chemotherapy of drug-resistant tumors. In this light, disclosing compounds with multitarget activity, combining MAO inhibition and antiproliferative effects, would strengthen the validity of the multitargeting approach in anticancer therapy.

The cytotoxicity data in Table 5 indicate that 10-indolyl THCP analogs **6a**–**c** are the most active among the tested compounds. They showed selectivity towards the MCF-7 line, with single-digit micromolar IC_50_s (4.80 ÷ 6.82 µM). Compounds **2a**–**b** and **3b**–**c** resulted in lower antitumor activity in MCF-7 cell line and poorly active at 50 μM concentration in SK-OV-3 cells.

Spearman’s rank analysis [15] for the pairs **6a**/CDDP/DXR, **6b**/CDDP/DXR and **6c**/CDDP/DXR provided very low values of Spearman indexes (*ρ* = −0.5), suggesting that the growth inhibition profile of the most active **6a**–**c** in all the examined tumor cell lines is different from both CDDP and DXR. Considering that the SK-OV-3 cell line is characterized by intrinsic resistance, the antiproliferative activity of compounds **6a**–**c** led us to hypothesize their use towards resistant lines and/or a synergistic action with drugs used in conventional therapies.

#### 2.2.3. Structure–Activity Relationships

The biological evaluation of the newly synthesized molecules provided us with SARs which may helpfully suggest and support future ‘hit-to-lead’ molecular optimization studies of 1*H*-chromeno[3,2-*c*]pyridine analogs as MAO isoform-selective inhibitors or MAO-targeted neuroprotectant MTDLs, in combination with molecular modeling results obtained by others on chromone-based natural and synthetic compounds [2,16]. From the SAR perspective, within the limits of the biological and physicochemical space explored, this ‘target-to-hit’ study proves that: (i) The 2,3-DHCP derivatives **2a**–**b** inhibit preferentially MAO A with IC_50_s of about 1 μM; (ii) the most potent MAO B-selective inhibitors are the 2-methyl 1,2,3,4-THCP-10-one derivatives **3a** and **3b** (IC_50_s 0.51 and 0.63 μM), which bear a phenylethynyl fragment at C1; **3a** achieved also IC_50_s of 7–8 μM against both ChEs; (iii) installing the 1*H*-indol-3-yl fragment on C10 of the starting compound **2a** did slightly improve the MAO B inhibition potency, and AChE as well, with a small effect of the 5-Br-indolyl on the activity of **6c**, which inhibited MAO B with a potency (IC_50_ 3.51 μM) close to that of pargyline; (iv) irrespective of their diastereisomerism, the cyclic hemiacetals **7** and **8** lose inhibition potency against the tested enzymes, likely owing to a loss of flatness compared to their more closely related compounds **2**.

In addition, the tumor growth inhibitory activity assayed in three cell lines (i.e., MCF-7, HCT116 and SK-OV-3) suggests that the 10-(1*H*-indol-3-yl)-bearing 2,3,4,10-THCP analogs **6a**–**c** are noteworthy. Based on molecular docking models and fluorescence quenching experiments, carried out by some of us on similar molecules [9], a propensity of compound **6** to bind DNA cannot be ruled out. However, **6c** exerted antiproliferative effects with IC_50_s < 10 μM, with a value of 11 μM measured toward the cisplatin-resistant ovarian tumor cells (SK-OV-3). Alongside a more in-depth mechanistic investigation and molecular optimization, the hit compound **6c** would also deserve to be tested against other tumor cell lines, trying to improve its delivery through suitable formulations [17].

## 3. Materials and Methods

### 3.1. Chemistry

#### 3.1.1. General Methods

Materials and general procedures. All reagents and solvents were purchased from Merck (Darmstadt, Germany), J.T. Baker (Phillipsburg, NJ, USA) or Sigma-Aldrich Chemical Co. (St. Louis, MO, USA) and, unless specified, used without further purification. The melting points (m.p.) of all the compounds were determined on a SMELTING POINT 10 apparatus in open capillaries (Bibby Sterilin Ltd., Stone, UK). IR spectra were recorded on an Infralum FT-801 FTIR spectrometer (ISP SB RAS, Novosibirsk, Russia). The samples were analyzed as KBr disk solids, and the most important frequencies in cm^−1^ are reported. ^1^H and ^13^C NMR spectra were recorded in chloroform-*d*_3_ (CDCl_3_) or dimethylsulfoxide-*d*_6_ (DMSO-*d*_6_) solutions at 25 °C, with a 600-MHz NMR spectrometer (JEOL Ltd., Tokyo, Japan). Peak positions were given in parts per million (ppm) referenced to the appropriate solvent residual peak, and signal multiplicities were collected by: s (singlet), d (doublet), t (triplet), q (quartet), dd (doublet of doublets), ddd (doublet of doublet of doublet), tt (triplet of triplets), br.s (broad singlet) and m (multiplet). MALDI mass spectra were recorded using a Bruker autoflex speed instrument operating in positive reflectron mode (Bremen, Germany). The data of **3a**, **7c** and **8a** were collected at room temperature using a STOE diffractometer Pilatus100K detector, focusing on mirror collimation Cu Kα (1.54086 Å) radiation, in rotation method mode. STOE X-AREA software was used for cell refinement and data reduction. Data collection and image processing were performed with X-Area 1.67 (STOE and Cie GmbH, Darmstadt, Germany, 2013). Intensity data were scaled with LANA (part of X-Area) to minimize the differences in intensities of symmetry equivalent reflections (multi-scan method). The structures were solved and refined with SHELX (Sheldrick, G.M. Acta Crystallogr. 2008, A64, 112–122.) program. The non-hydrogen atoms were refined by using the anisotropic full matrix least-square procedure. Molecular geometry calculations were performed with the SHELX program, and the molecular graphics were prepared by using DIAMOND software (Brandenburg, K. DIAMOND, Release 2.1d; Crystal Impact GbR: Bonn, Germany, 2000).

#### 3.1.2. Synthesis of Tetrahydro- and Dihydrochromeno[3,2-*c*]pyridines **1** and **2**

The synthesis of compounds **1a**–**d** was described in the article [5,6,7].

The synthesis of 2-alkyl-2,3-dihydro-1*H*-chromeno[3,2-*c*]pyridines **2**.

Compound **1**, *para*-toluene sulfonic acid (20 mol%) and hydroquinone (10 mol%) were placed into a 100 mL flask and *o*-xylene (50 mL) was added. The mixture was boiled for 6 h with a Dean-Stark Moisture Trap, until the calculated amount of water was released. The solvent was evaporated, and the product was obtained by crystallization from diethyl ether in the form of a grayish-yellow powder, filtered on a glass filter and dried in air.

**8-Chloro-2,3-dihydro-2-methyl-1*H*-chromeno[3,2-c]pyridine (2a).** Yield 45%, brown crystals, m.p. = 133–134 °C. ^1^H NMR (600 MHz, CDCl_3_) δ (ppm): 2.40 (s, 3H), 3.18 (d, *J* = 4.1 Hz, 2H), 3.24 (d, *J* = 1.5 Hz, 2H), 5.08 (dt, *J* = 4.2, 1.8 Hz, 1H), 5.99 (s, 1H), 6.70 (d, *J* = 8.6 Hz, 1H), 6.89 (d, *J* = 2.5 Hz, 1H), 7.01 (dd, *J* = 8.6, 2.5 Hz, 1H). ^13^C NMR (151 MHz, CDCl_3_) δ (ppm): 44.7, 53.7, 56.9, 98.5, 115.9, 117.0, 123.0, 125.7, 126.9, 128.4, 129.5, 147.6, 151.6. HRMS (MALDI+) *m*/*z* calcd for C_13_H_12_ClNO in form of [M + H]^+^ ion 234.0686, found: 234,0693.

**8-Bromo-2,3-dihydro-2-methyl-1*H*-chromeno[3,2-*c*]pyridine (2b).** Yield 38%, brown crystals, m.p. = 137–138 °C. ^1^H NMR (600 MHz, CDCl_3_) δ (ppm): 2.40 (s, 3H), 3.18 (d, *J* = 4.1 Hz, 2H), 3.25 (d, *J* = 1.4 Hz, 2H), 5.08 (td, *J* = 4.2, 1.8 Hz, 1H), 5.98 (s, 1H), 6.64 (d, *J* = 8.6 Hz, 1H), 7.03 (d, *J* = 2.4 Hz, 1H), 7.15 (dd, *J* = 8.6, 2.4 Hz, 1H). ^13^C NMR (151 MHz, CDC_l3_) δ (ppm): 44.8, 53.7, 56.9, 98.6, 114.2, 116.5, 116.9, 123.6, 128.6, 129.6, 131.3, 147.6, 152.2. HRMS (MALDI+) *m*/*z* calcd for C_13_H_12_BrNO in form of [M + H]^+^ ion 278.0181, found: 278,0193.

#### 3.1.3. Synthesis of 2-Alkyl-1-(ethinyl)-1*H*-chromeno[3,2-*c*]pyridine-10(2*H*)-ones (**3a**–**e**)

A solution of compound **1** (0.4 g, 0.00163 mol) in THF was cooled to 0 °C, and 1.2 equiv. of DIAD was added and stirred at room temperature for 1 h. Then, it was cooled again to 0 °C, 3 equiv. of phenylacetylene and a CuI (20 mol %) catalyst were added. The reaction was conducted at r.t. and constant stirring, and reaction was monitored by TLC. The solvent was evaporated, and the product was purified by column chromatography.

**2-Methyl-1-(phenylethinyl)-1*H*-chromeno[3,2-*c*]pyridine-10(2*H*)-one (3a)**. Yield 44%, yellow crystals, m.p. = 140–141 °C. ^1^H NMR (600 MHz, CDCl_3_) δ (ppm): 3.19 (s, 3H), 5.11 (d, *J* = 7.3 Hz, 1H), 5.94 (d, *J* = 1.2 Hz, 1H), 6.62 (dd, *J* = 7.3, 1.3 Hz, 1H), 7.22–7.26 (m, 3H), 7.29–7.36 (m, 2H), 7.37–7.42 (m, 2H), 7.55 (ddd, *J* = 8.7, 7.1, 1.7 Hz, 1H), 8.19 (dd, *J* = 7.9, 1.6 Hz, 1H). IR spectra (KBr), cm^−1^: 2250.8 (-C≡C-). HRMS (MALDI+) *m*/*z* calcd for C_21_H_15_NO_2_ in form of [M + H]^+^ ion 314.1181, found: 314.1195. Crystals suitable for X-ray crystallography were obtained by slow crystallization of a solution in methanol.

**6-Metoxy-2-methyl-1-(phenylethinyl)-1*H*-chromeno[3,2-*c*]pyridine-10(2*H*)-one (3b).** Yield 38%, yellow crystals, m.p. = 169–170 °C. ^1^H NMR (600 MHz, CDCl_3_) δ (ppm): 3.19 (s, 3H), 3.97 (s, 3H), 5.22 (d, *J* = 7.3 Hz, 1H), 5.93 (d, *J* = 1.2 Hz, 1H), 6.62 (dd, *J* = 7.3, 1.2 Hz, 1H), 7.09 (dd, *J* = 8.0, 1.5 Hz, 1H), 7.22–7.26 (m, 4H), 7.37–7.40 (m, 2H), 7.76 (dd, *J* = 8.0, 1.5 Hz, 1H). ^13^C NMR (151 MHz, CDCl_3_) δ (ppm): 41.2, 49.0, 56.5, 84.2, 85.7, 88.5, 102.6, 113.8, 116.9, 122.7, 124.0, 125.7, 128.2 (2C), 128.4, 132.2 (2C), 145.6, 147.5, 148.5, 161.3, 173.3. HRMS (MALDI+) *m*/*z* calcd for C_22_H_17_NO_3_ in form of [M + H]+ ion 344.1287, found: 344.1273.


**2-Ethyl-1-(3,3,3-trifluoroprop-1-yn-1-yl)-1,2-dihydro-10*H*-chromeno[3,2-*c*]pyridin-10-one (3c).**


A solution of 2-ethyl-1*H*-chromeno[3,2-*c*]pyridin-10(2*H*)-one 0.4 g (1.74 mmol) in THF was cooled to 0 °C, and 3 equiv. excess of DIAD was added and stirred at r.t. for 1 h. Then, the reaction mixture was cooled to −70 °C, the CuI catalyst was added and a two-fold excess of gaseous alkyne was condensed to the mixture. The ampoule with the reaction mixture was sealed and placed in a protective metal cylinder. The reaction mixture was kept at 6 °C for one day, and then 5 days at room temperature. The reaction was controlled by TLC using ethyl acetate/n-hexane 1:1 *v*/*v* as eluent, Silufol. The solvent was evaporated under a vacuum. The reaction product was purified by column chromatography for SiO_2_, eluent- ethylacetate/n-hexane 1:3 *v*/*v*. Yellow crystals, 23% yield, m.p. = 96–97 °C. ^1^H NMR (600 MHz, CDCl_3_) δ (ppm): 1.34 (t, *J* = 7.3 Hz, 3H), 3.35 (dt, *J* = 14.4, 7.3 Hz, 1H), 3.41–3.53 (m, 1H), 5.18 (d, *J* = 7.3 Hz, 1H), 5.93–6.03 (m, 1H), 6.70 (dd, *J* = 7.3, 1.3 Hz, 1H), 7.30–7.40 (m, 2H), 7.58 (ddd, *J* = 8.6, 7.1, 1.7 Hz, 1H), 8.18 (dd, *J* = 7.9, 1.7 Hz, 1H). ^13^C NMR (151 MHz, CDCl_3_) δ (ppm): 13.9, 46.1, 49.0, 70.8, 84.5, 89.3, 100.4, 114.9, 117.7, 124.3, 124.8, 125.6, 133.1, 146.2, 155.4, 161.8, 173.2. IR (KBr), cm^−1^: 2270.8 (-C≡C-). HRMS (MALDI+) *m*/*z* calcd for C_17_H_12_F_3_NO_2_ in form of [M + H]^+^ ion 319.0820, found: 319.0832.

**6-Ethoxy-2-isopropyl-1-(phenylethynyl)-1,2-dihydro-10*H*-chromeno[3,2-*c*]pyridin-10-one (3d).** Green crystals, 20% yield, m.p. = 157–158 °C. ^1^H NMR (600 MHz, CDCl_3_) δ (ppm): 1.43 (d, *J* = 6.56, 6H), 1.53 (t, *J* = 7.06, 3H), 4.19 (q, *J* = 7.06, 2H), 5.01 (m, 1H), 5.59 (s, 1H), 6.57 (d, *J* = 7,06, 1H), 6.77 (d, *J* = 8,07, 1H), 6.94 (t, *J* = 7.57, 1H), 7.17 (t, *J* = 7.57, 2H) 7.33–7.40 (m, 4H), 7.44 (d, *J* = 8.07, 1H). ^13^C NMR (151 MHz, CDCl_3_) δ (ppm): 193.2, 158.3, 148.3, 142.1, 140.6, 136.6, 133.0, 132.4, 131.8, 130.4, 129.6, 128.1, 128.0, 125.4, 123.5, 120.3, 117.1, 112.9, 111.8, 109.3, 98.6, 65.3, 54.3, 30.0, 22.3 (2C), 15.2. IR (KBr), cm^−1^: 2250.8 (-C≡C-). HRMS (MALDI+) *m*/*z* calcd for C_25_H_23_NO_3_ in form of [M + H]^+^ ion 386.1756, found: 386.1770.

#### 3.1.4. Synthesis of 8-Bromo-2-methyl-10-(1-methyl-1*H*-pyrrole-2-yl)-2,3,4,10-tetrahydro-1*H*-chromeno[3,2-c]pyridine (**4**)

Chromenopyridine **2b** (0.2 g, 0.85 mmol) and pyrrole (1.5 equiv.) were dissolved in trifluoroethanol (4 mL) and heated in a microwave oven at 150 °C for 1.5 h (3 times for 30 min). The solvent had evaporated. The product was isolated by column chromatography. Red oil, 52% yield. ^1^H NMR (600 MHz, CDCl_3_) δ (ppm): 2.34 (s, 3H), 2.41–2.46 (m, 2H), 2.54–2.63 (m, 2H), 2.69–2.74 (m, 1H), 2.80 (d, *J* = 14.8 Hz, 1H), 3.30 (s, 3H), 4.67 (s, 1H), 6.02 (t, *J* = 3.1 Hz, 1H), 6.04–6.06 (m, 1H), 6.49 (t, *J* = 2.3 Hz, 1H), 6.79 (d, *J* = 8.7 Hz, 1H), 7.04 (d, *J* = 2.5 Hz, 1H), 7.22 (dd, *J* = 8.7, 2.5 Hz, 1H). HRMS (MALDI+) *m*/*z* calcd for C_18_H_19_BrN_2_O in form of [M + H]^+^ ion 359.0759, found: 359.0772.

#### 3.1.5. Synthesis of 8-Bromo-2-methyl-10-(nitromethyl)-2,3,4,10-tetrahydro-1*H*-chromeno[3,2-*c*]pyridine (**5**)

Chromenopyridine **2b** (0.2 g, 0.00085 mol), nitromethane (3 equiv.) and triethylamine (2 equiv.) were dissolved in trifluoroethanol (4 mL) and heated in a microwave oven at 150 °C for 1.5 h (3 times for 30 min). The solvent was evaporated, and the product was isolated by column chromatography. Beige crystals, 56% yield, m.p. 140–142 °C. ^1^H NMR (600 MHz, CDCl_3_) δ (ppm): 2.34–2.40 (m, 1H), 2.40–2.45 (m, 4H), 2.52–2.57 (m, 1H), 2.73–2.79 (m, 1H), 2.96–3.03 (m, 2H), 4.02 (t, *J* = 6.1 Hz, 1H), 4.41 (dd, *J* = 12.1, 6.8 Hz, 1H), 4.50 (dd, *J* = 12.1, 5.2 Hz, 1H), 6.86 (d, *J* = 8.8 Hz, 1H), 7.23 (d, *J* = 2.3 Hz, 1H), 7.33 (dd, *J* = 8.6, 2.3 Hz, 1H). ^13^C NMR (151 MHz, CDCl_3_) δ (ppm): 27.4, 37.3, 45.5, 52.0, 55.2, 80.1, 102.1, 115.9, 118.6, 121.6, 130.7, 131.9, 147.0, 150.9. HRMS (MALDI+) *m*/*z* calcd for C_14_H_15_BrN_2_O_3_ in form of [M + H]^+^ ion 339.0344, found: 339.0357.

#### 3.1.6. Synthesis of 8-Chloro-10-(1*H*-indol-3-yl)-2-methyl-2,3,4,10-tetrahydro-1*H*-chromeno[3,2-c]pyridines **6a**–**c**

Compound **2a** (0.2 g, 0.85 mmol) and indole (0.85 mmol) were dissolved in trifluoroethanol (4 mL) and heated in a microwave oven at 150 °C for 1.5 h (3 times for 30 min). The solvent had evaporated. The product was isolated by column chromatography in the form of yellow-brown foamed oil.

**8-Chloro-10-(1*H*-indole-3-yl)-2-methyl-2,3,4,10-tetrahydro-1*H*-chromeno[3,2-*c*]pyridine (6a).** Beige foamed oil, yield 21%. ^1^H NMR (600 MHz, CDCl_3_) δ (ppm): 2.41 (s, 3H), 2.58 (d, *J* = 16.5 Hz, 1H), 2.65–2.75 (m, 1H), 2.74–2.89 (m, 2H), 2.96–3.05 (m, 1H), 3.13 (d, *J* = 14.7 Hz, 1H), 4.67 (s, 1H), 6.90 (d, *J* = 8.7 Hz, 1H), 6.94 (s, 1H), 7.00–7.06 (m, 2H), 7.10 (s, 1H), 7.15 (t, *J* = 7.6 Hz, 1H), 7.36 (d, *J* = 8.0 Hz, 1H), 7.42 (d, *J* = 8.0 Hz, 1H), 8.49 (s, 1H). ^13^C NMR (151 MHz, CDCl_3_) δ (ppm): 27.2, 34.8, 45.3, 52.2, 55.4, 106.5, 111.4, 117.6, 119.1, 119.3, 119.9, 122.4, 122.5, 125.4, 126.2, 127.6, 127.6, 129.4, 136.9, 142.3, 149.3. HRMS (MALDI+) *m*/*z* calcd for C_21_H_19_ClN_2_O in form of [M + H]^+^ ion 351.1264, found: 351,1250.

**8-Chloro-10-(5-metoxy-1*H*-indole-3-yl)-2-methyl-2,3,4,10-tetrahydro-1*H*-chromeno[3,2-*c*]pyridine (6b).** Beige foamed oil, yield 20%. ^1^H NMR (600 MHz, CDCl_3_) δ (ppm): 2.27 (s, 2H), 2.40–2.49 (m, 2H), 2.44–2.59 (m, 3H), 2.65 (d, *J* = 14.7 Hz, 1H), 2.68–2.77 (m, 1H), 2.85 (d, *J* = 14.7 Hz, 1H), 3.76 (s, 3H), 4.63 (s, 1H), 6.82 (dd, *J* = 8.8, 2.4 Hz, 1H), 6.89 (d, *J* = 8.8 Hz, 1H), 6.93 (d, *J* = 2.5 Hz, 1H), 6.97 (dd, *J* = 2.5, 0.8 Hz, 1H), 6.99–7.07 (m, 2H), 7.22 (d, *J* = 8.8 Hz, 1H), 8.14 (s, 1H). ^13^C NMR (151 MHz, CDCl_3_) δ (ppm): 27.2, 34.8, 45.3, 52.2, 55.4, 55.8, 101.2, 105.3, 112.0, 112.2, 117.4, 118.7, 123.1, 125.3, 126.6, 127.5, 127.6, 129.4, 132.0, 142.2, 149.4, 154.0. HRMS (MALDI+) *m*/*z* calcd for C_22_H_21_ClN_2_O_2_ in form of [M + H]^+^ ion 381.1370, found: 381,1391.

**8-Chloro-10-(5-bromo-1*H*-indole-3-yl)-2-methyl-2,3,4,10-tetrahydro-1*H*-chromeno[3,2-*c*]pyridine (6c)**. Beige foamed oil, yield 51%. ^1^H NMR (600 MHz, CDCl_3_) δ (ppm): 2.33 (3H, s), 2.53–2.58 (2H, m), 2.69–2.76 (2H, m), 2.84–2.89 (1H, m), 2.96 (1H, d, *J* = 14.9 Hz), 4.59 (1H, s), 6.87–6.92 (2H, m), 7.02 (1H, s), 7.05 (1H, dd, *J* = 8.6, 1.9 Hz), 7.19 (2H, s), 7.53 (1H, s), 8.99 (1H, s).^13^C NMR (150 MHz, CDCl_3_), δ (ppm): 25.27, 34.12, 43.66, 51.21, 53.81, 113.18, 113.27, 117.31, 117.78, 121.20, 124.33, 124.41, 125.33, 125.41, 127.56, 128.03, 128.21, 129.13, 135.50, 141.88, 148.69. HRMS (MALDI+) *m*/*z* calcd for C_21_H_18_ClBrN_2_O in form of [M + H]^+^ ion 429.0369, found: 429.0382.

#### 3.1.7. Synthesis of (4aR*,10S*,10aR*)-2-Alkyl-1,2,3,4,10,10a-hexahydro-4a*H*-chromeno[3,2-c]pyridine-4a,10-diol **7a**–**f**

Corresponding salicylic aldehyde (2.6 mmol) and *N*-alkylpiperidone (2.6 mmol) and *L*-proline (10 mol%) were dissolved in ethanol (~10 mL). The mixture was heated and stirred at 75 °C for one day. The crystals that fell out after cooling were filtered and dried in air. In some cases, an additional portion of the substance was isolated from the liquor after partial evaporation of the solvent.

**(4aR*,10S*,10aR*)-2-Methyl-1,2,3,4,10,10a-hexahydro-4a*H*-chromeno[3,2-*c*]pyridine-4a,10-diol (7a)**. White crystals, yield 79%, m.p. = 192–194 °C. ^1^H NMR (DMSO-*d*_6_, 600 MHz) δ (ppm): 1.78–1.91 (m, 3H), 1.94 (t, *J* = 10.9 Hz, 1H), 2.14–2.20 (m, 1H), 2.22 (s, 3H), 2.67–2.72 (m, 1H), 3.00 (dd, *J* = 10.7, 2.5 Hz, 1H), 4.33 (d, *J* = 10.7 Hz, 1H), 5.21 (s, 1H), 6.30 (s, 1H), 6.68–6.71 (m, 1H), 6.87–6.90 (m, 1H), 7.11 (td, *J* = 7.9, 1.6 Hz, 1H), 7.41 (dt, *J* = 7.9, 1.6 Hz, 1H). ^13^C NMR (151 MHz, DMSO-*d*_6_) 36.7, 45.6, 45.7, 52.2, 55.3, 63.3, 96.8, 116.0, 120.0, 127.2, 127.9, 128.0, 151.5. IR (KBr), cm^−1^: 2790–3100 (OH). HRMS (MALDI+) *m*/*z* calcd for C_13_H_17_NO_3_ in form of [M + H]^+^ ion 236.1287, found: 236.1299.

**(4aR*,10S*,10aR*)-8-Chloro-2-methyl-1,2,3,4,10,10a-hexahydro-4a*H*-chromeno[3,2-*c*]pyridine-4a,10-diol (7b)**. White crystals, yield 84%, m.p. = 172–173 °C. ^1^H NMR (600 MHz, DMSO-*d*_6_) δ (ppm): 1.78–1.86 (m, 2H), 1.89–1.97 (m, 2H), 2.13–2.17 (m, 1H), 2.22 (s, 3H), 2.68–2.73 (m, 1H), 2.99 (ddd, *J* = 10.8, 4.1, 1.6 Hz, 1H), 4.32 (dd, *J* = 10.8, 7.9 Hz, 1H), 5.41 (d, *J* = 8.0 Hz, 1H), 6.46 (d, *J* = 1.7 Hz, 1H), 6.73 (d, *J* = 8.7 Hz, 1H), 7.13–7.16 (m, 1H), 7.39 (dd, *J* = 2.8, 1.0 Hz, 1H). ^13^C NMR (DMSO-*d*_6_, 151 MHz) δ (ppm): 36.5, 45.1, 45.6, 52.1, 55.2, 63.2, 97.3, 118.0, 123.9, 126.8, 127.8, 130.1, 150.4. IR (KBr), cm^−1^: 2720–3300 (OH). HRMS (MALDI+) *m*/*z* calcd for C_13_H_16_ClNO_3_ in form of [M + H]^+^ ion 270.0897, found: 270.0878.

**(4aR*,10S*,10aR*)-8-Bromo-2-methyl-1,2,3,4,10,10a-hexahydro-4a*H*-chromeno[3,2-*c*]pyridine-4a,10-diol (7c)**. White crystals, yield 86%, m.p. = 160–161 °C. ^1^H NMR (600 MHz, DMSO-*d*_6_) δ (ppm): 1.77–1.86 (m, 2H), 1.88–1.92 (m, 1H), 1.93–1.96 (m, 1H), 2.13–2.16 (m, 1H), 2.22 (s, 3H), 2.68–2.73 (m, 1H), 2.99 (ddd, *J* = 10.6, 4.1, 1.6 Hz, 1H), 4.33 (dd, *J* = 10.9, 7.9 Hz, 1H), 5.41 (d, *J* = 8.1 Hz, 1H), 6.47 (d, *J* = 1.7 Hz, 1H), 6.69 (d, *J* = 8.6 Hz, 1H), 7.25–7.29 (m, 1H), 7.51 (dd, *J* = 2.6, 1.0 Hz, 1H). ^13^C NMR (DMSO-*d*_6_, 151 MHz) δ (ppm): 36.5, 45.1, 45.6, 52.1, 55.1, 63.1, 97.3, 111.6, 118.5, 129.7, 130.6, 130.6, 150.9. IR (KBr), cm^−1^: 2723–3329 (OH). HRMS (MALDI+) *m*/*z* calcd for C_13_H_16_BrNO_3_ in form of [M + H]^+^ ion 314.0391, found: 314.0407. Crystals suitable for X-ray crystallography were obtained by slow crystallization of a solution in methanol.

**(4aR*,10S*,10aR*)-6-Methoxy-2-methyl-1,2,3,4,10,10a-hexahydro-4a*H*-chromeno[3,2-*c*]pyridine-4a,10-diol (7d).** White crystals, yield 74%, m.p. = 134–135 °C. ^1^H NMR (600 MHz, DMSO-*d*_6_) δ (ppm): 1.78–1.87 (m, 2H), 1.90–1.93 (m, 1H), 1.93–1.96 (m, 1H), 2.12–2.16 (m, 1H), 2.22 (s, 3H), 2.67–2.73 (m, 1H), 2.97–3.02 (m, 1H), 3.71 (s, 3H), 4.31 (d, *J* = 10.8 Hz, 1H), 5.17 (s, 1H), 6.32 (s, 1H), 6.80–6.82 (m, 2H), 7.00–7.02 (m, 1H). ^13^C NMR (DMSO-*d*_6_, 151 MHz) δ (ppm): 36.8, 45.4, 45.7, 52.2, 55.3, 55.3, 63.4, 96.7, 110.4, 118.9, 119.4, 128.4, 140.9, 147.6. HRMS (MALDI+) *m*/*z* calcd for C_14_H_19_NO_4_ in form of [M + H]^+^ ion 266.1392, found: 266.1405.

**(4aR*,10S*,10aR*)-6-Ethoxy-2-methyl-1,2,3,4,10,10a-hexahydro-4a*H*-chromeno[3,2-*c*]pyridine-4a,10-diol (7e).** White crystals, yield 71%, m.p. = 168–169 °C. ^1^H NMR (600 MHz, DMSO-*d*_6_) δ (ppm): 1.30 (t, *J* = 6.9 Hz, 3H), 1.74–1.91 (m, 2H), 1.91–1.95 (m, 2H), 2.10–2.19 (m, 1H), 2.22 (s, 3H), 2.66–2.72 (m, 1H), 2.95–3.03 (m, 1H), 3.92–4.00 (m, 2H), 4.31 (dd, *J* = 10.8, 8.0 Hz, 1H), 5.19 (d, *J* = 8.2 Hz, 1H), 6.35 (d, *J* = 1.7 Hz, 1H), 6.79 (d, *J* = 1.9 Hz, 1H), 6.80 (s, 1H), 6.96–7.03 (m, 1H). ^13^C NMR (DMSO-*d*_6_, 151 MHz) δ (ppm): 14.9, 36.8, 45.4, 45.7, 52.3, 55.4, 63.5, 63.7, 96.7, 111.9, 119.0, 119.4, 128.6, 141.2, 146.8. IR (KBr), cm^−1^: 3475, 2773–3118 (OH). HRMS (MALDI+) *m*/*z* calcd for C_15_H_21_NO_4_ in form of [M + H]^+^ ion 280.1549, found: 280.1536.

**(4aR*,10S*,10aR*)-2-Methyl-8-nitro-1,2,3,4,10,10a-hexahydro-4a*H*-chromeno[3,2-*c*]pyridine-4a,10-diol (7f).** Orange crystals, yield 96%, m.p. = 158–159 °C. ^1^H NMR (600 MHz, DMSO-*d*_6_) δ (ppm): 1.86–1.94 (m, 2H), 1.99–2.05 (m, 2H), 2.20–2.24 (m, 1H), 2.26 (s, 3H), 2.72–2.81 (m, 1H), 3.03–3.09 (m, 1H), 4.43 (d, *J* = 11.1 Hz, 1H), 5.75 (br.s, 1H), 6.90 (br.s, 1H), 6.95 (dd, *J* = 9.0, 1.4 Hz, 1H), 8.04 (dd, *J* = 9.0, 2.9 Hz, 1H), 8.33 (dd, *J* = 2.9, 1.1 Hz, 1H). ^13^C NMR (DMSO-*d*_6_, 151 MHz) δ (ppm): 36.6, 45.3, 45.9, 52.5, 55.5, 63.3, 99.3, 117.8, 124.0, 124.7, 129.6, 141.3, 157.9. IR (KBr), cm^−1^: 2670–3263 (OH), 1339, 1511 (NO_2_). HRMS (MALDI+) *m*/*z* calcd for C_13_H_16_N_2_O_5_ in form of [M + H]^+^ ion 281.1137, found: 281.1149.

#### 3.1.8. Synthesis of (4aR*,10S*,10aS*)-2-Alkyl-1,2,3,4,10,10a-hexahydro-4a*H*-chromeno[3,2-c]pyridine-4a,10-diols **8a**–**e**

The corresponding salicylic aldehyde (0.0026 mol) and *N*-alkylpiperidone (0.0026 mol) were dissolved in ethanol (~10 mL). The mixture was heated and stirred at 75 °C for one day. The crystals that fell out after cooling were filtered and dried in air. In some cases, an additional portion of the substance was isolated from the liquor after partial evaporation of the solvent.

**(4aR*,10S*,10aS*)-2-Methyl-1,2,3,4,10,10a-hexahydro-4a*H*-chromeno[3,2-*c*]pyridine-4a,10-diol (8a).** White crystals, yield 92%, m.p. = 187–188 °C. ^1^H NMR (600 MHz, DMSO-*d*_6_) δ (ppm): 1.80–1.87 (m, 2H), 1.87–1.92 (m, 1H), 1.93–1.98 (m, 1H), 2.12–2.23 (m, 1H), 2.23 (s, 3H), 2.68–2.73 (m, 1H), 3.01 (ddd, *J* = 10.6, 4.0, 1.6 Hz, 1H), 4.34 (dd, *J* = 10.9, 7.9 Hz, 1H), 5.22 (d, *J* = 8.2 Hz, 1H), 6.30 (d, *J* = 1.8 Hz, 1H), 6.70 (dd, *J* = 8.1, 1.2 Hz, 1H), 6.89 (td, *J* = 7.5, 1.3 Hz, 1H), 7.11 (td, *J* = 7.7, 1.8 Hz, 1H), 7.42 (d, *J* = 7.7 Hz, 1H). ^13^C NMR (DMSO-*d*_6_, 151 MHz) δ (ppm): 36.7, 45.6, 45.7, 52.2, 55.3, 63.3, 96.8, 116.0, 120.0, 127.2, 127.9, 128.0, 151.5. IR (KBr), cm^−1^: 2720–3300 (OH). HRMS (MALDI+) *m*/*z* calcd for C_13_H_17_NO_3_ in form of [M + H]^+^ ion 236.1287, found: 236.1300. Crystals suitable for X-ray crystallography were obtained by slow crystallization of a solution in methanol.

**(4aR*,10S*,10aS*)-8-Chloro-2-methyl-1,2,3,4,10,10a-hexahydro-4a*H*-chromeno[3,2-*c*]pyridine-4a,10-diol (8b).** White crystals, yield 87%, m.p. = 167–168 °C. ^1^H NMR (600 MHz, DMSO-*d*_6_) δ (ppm): 1.79–1.87 (m, 2H), 1.89–1.97 (m, 2H), 2.14–2.17 (m, 1H), 2.22 (s, 3H), 2.68–2.73 (m, 1H), 3.00 (ddd, *J* = 10.6, 4.1, 1.6 Hz, 1H), 4.32 (dd, *J* = 10.9, 7.9 Hz, 1H), 5.42 (d, *J* = 8.0 Hz, 1H), 6.46 (d, *J* = 1.7 Hz, 1H), 6.73 (d, *J* = 8.6 Hz, 1H), 7.15 (ddd, *J* = 8.6, 2.7, 0.7 Hz, 1H), 7.39 (dd, *J* = 2.7, 1.0 Hz, 1H). ^13^C NMR (DMSO-*d*_6_, 151 MHz) δ (ppm): 36.5, 45.1, 45.6, 52.1, 55.2, 63.2, 97.3, 118.0, 123.9, 126.8, 127.8, 130.1, 150.4. IR (KBr), cm^−1^: 2720–3300 (OH). HRMS (MALDI+) *m*/*z* calcd for C_13_H_16_ClNO_3_ in form of [M + H]^+^ ion 270.0897, found: 270.0905.

**(4aR*,10S*,10aS*)-8-Bromo-2-methyl-1,2,3,4,10,10a-hexahydro-4a*H*-chromeno[3,2-*c*]pyridine-4a,10-diol (8c)**. White crystals, yield 90%, m.p. = 173–174 °C. ^1^H NMR (600 MHz, DMSO-*d*_6_) δ (ppm): 1.78–1.87 (m, 2H), 1.87–1.92 (m, 1H), 1.92–1.97 (m, 1H), 2.13–2.17 (m, 1H), 2.22 (s, 3H), 2.68–2.73 (m, 1H), 2.99 (ddd, *J* = 10.7, 4.1, 1.6 Hz, 1H), 4.33 (dd, *J* = 10.9, 7.9 Hz, 1H), 5.41 (d, *J* = 8.0 Hz, 1H), 6.47 (d, *J* = 1.7 Hz, 1H), 6.69 (d, *J* = 8.6 Hz, 1H), 7.27 (dd, *J* = 8.6, 2.6 Hz, 1H), 7.52 (dd, *J* = 2.6, 1.0 Hz, 1H). ^13^C NMR (DMSO-*d*_6_, 151 MHz) δ (ppm): 37.0, 45.7, 46.1, 52.6, 55.7, 63.6, 97.8, 112.1, 119.0, 130.2, 131.1, 131.2, 151.4. IR (KBr), cm^−1^: 2723–3329 (OH). HRMS (MALDI+) *m*/*z* calcd for C_13_H_16_BrNO_3_ in form of [M + H]^+^ ion 314.0391, found: 314.0401.

**(4aR*,10S*,10aS*)-6-Methoxy-2-methyl-1,2,3,4,10,10a-hexahydro-4a*H*-chromeno[3,2-*c*]pyridine-4a,10-diol (8d)** Light-pink crystals, yield 83%, m.p. = 148–149 °C. ^1^H NMR (600 MHz, DMSO-*d*_6_) δ (ppm): 1.79–1.87 (m, 2H), 1.90–1.96 (m, 2H), 2.14–2.18 (m, 1H), 2.22 (s, 3H), 2.67–2.72 (m, 1H), 3.00 (ddd, *J* = 10.6, 4.1, 1.6 Hz, 1H), 3.71 (s, 3H), 4.31 (d, *J* = 10.8 Hz, 1H), 5.17 (s, 1H), 6.32 (s, 1H), 6.80–6.82 (m, 2H), 7.00–7.02 (m, 1H). ^13^C NMR (DMSO-*d*_6_, 151 MHz) δ (ppm): 37.3, 45.9, 46.2, 52.7, 55.8, 55.9, 63.9, 97.2, 110.9, 119.4, 119.9, 128.9, 141.4, 148.1. HRMS (MALDI+) *m*/*z* calcd for C_14_H_19_NO_4_ in form of [M+H]^+^ ion 266.1392, found: 266.1407.

**(4aR*,10S*,10aS*)-2-Benzyl-1,2,3,4,10,10a-hexahydro-4a*H*-chromeno[3,2-*c*]pyridine-4a,10-diol (8e).** White crystals, yield 62%, m.p. 141–142 °C. ^1^H NMR (600 MHz, DMSO-*d*_6_) δ (ppm): 1.79–1.86 (m, 2H), 1.89–1.92 (m, 1H), 1.99–2.02 (m, 1H), 2.23–2.30 (m, 1H), 2.73–2.79 (m, 1H), 3.09 (ddd, *J* = 10.7, 4.1, 1.6 Hz, 1H), 3.49–3.60 (m, 2H), 4.31 (d, *J* = 10.9 Hz, 1H), 5.19 (s, 1H), 6.33 (s, 1H), 6.69 (dd, *J* = 8.1, 1.2 Hz, 1H), 6.88 (td, *J* = 7.4, 1.2 Hz, 1H), 7.08–7.12 (m, 1H), 7.25–7.33 (m, 5H), 7.37–7.40 (m, 1H). ^13^C NMR (DMSO-*d*_6_, 151 MHz) δ (ppm): 36.7, 45.6, 49.9, 52.9, 61.9, 63.4, 97.1, 116.0, 120.0, 126.9 (2C), 127.2, 127.9, 128.0, 128.2 (2C), 128.9, 138.5, 151.5. IR (KBr), cm^−1^: 2820–3357 (OH). HRMS (MALDI+) *m*/*z* calcd for C_19_H_21_NO_3_ in form of [M + H]^+^ ion 312.1600, found: 312.1590.

All ^1^H NMR and ^13^C NMR spectra presented in the Appendix A.

### 3.2. Biological Assays

#### 3.2.1. Inhibition of Monoamine Oxidases and Cholinesterases

Human isoforms of MAOs (from baculovirus-infected insect cells) and ChEs (human recombinant AChE and BChE from human serum), purchased from Sigma Aldrich (Milan, Italy), were used for inhibition assays. Experiments were performed in 96-well plates (Greiner Bio-One, Kremsmünster, Austria) on the Infinite M1000 Pro plate reader (Tecan, Cernusco s.N., Italy), using already published protocols [5,18,19]. Inhibition data and constants (IC_50_s and *K*_i_s) were calculated with Prism (version 5.01 for Windows; GraphPad Software, San Diego, CA, USA).

In MAOs’ inhibition assays, each test compound, at 10 μM concentration, was preincubated for 20 min at 37 °C with 50 μM kynuramine as the substrate in 0.1 M phosphate buffer solution (PBS) pH 8.0 made 0.39 osmolar with KCl. After the addition of human recombinant MAO A (250 U/mg) or MAO B (59 U/mg) and a further 30 min of incubation, NaOH was added, and the fluorescence read at 310/400 nm excitation/emission wavelength. For compounds achieving at least 60% inhibition of MAO at 10 μM concentration, seven scalar concentrations of each inhibitor were tested and the concentration producing 50% inhibition of the MAO activity (IC_50_) was calculated by nonlinear regression. IC_50_ is expressed as mean ± SD of three independent measurements, each one performed in duplicate. For the kinetic study on the inhibition mechanism of MAO B, three diverse scalar concentrations of the inhibitor and seven concentrations of kynuramine were used.

The inhibition of human recombinant AChE or BChE from human serum was determined by applying Ellman’s spectrophotometric method as described in previously reported protocols [18,19]. The AChE activity was determined in an assay solution containing AChE (0.09 U/mL), 5,5′-dithiobis(2-nitrobenzoic acid) (i.e., the Ellman’s reagent, 0.33 mM), the test compound (10 μM concentration, or seven scalar concentrations for compounds achieving > 60% enzyme inhibition at 10 μM), in 0.1 M PBS pH 8.0. After 20 min incubation at 25 °C, the substrate acetylthiocholine iodide (5 μM) was added, and its hydrolysis rates were monitored for 5.0 min at 412 nm. The BChE inhibitory activity was similarly determined by using BChE (0.09 U/mL) and butyrylthiocholine iodide (5 μM) as the substrate. IC_50_ value, determined by the nonlinear regression method ‘log[inhibitor] vs. response’, or the % inhibition at 10 μM, is expressed as the mean ± SD of three independent measurements, each one performed in duplicate.

The IC_50_ values, Michaelis–Menten curve fitting and inhibition constant (*K*_i_) were calculated by nonlinear regression, using Prism software.

#### 3.2.2. Cell Viability Assays

The SK-OV-3 ovarian cancer cell line, MCF-7 breast cancer line and HCT-116 colon cancer cell line were obtained from the National Cancer Institute, Biological Testing Branch (Frederick, MD, USA), and maintained in the logarithmic phase at 37 °C in a 5% CO_2_ humidified air in RPMI 1640 medium supplemented with 10% fetal calf serum, 2 mM glutamine, penicillin (100 U/mL) and streptomycin (0.1 mg/mL).

The growth inhibitory effects of compounds under investigation were compared to those of cisplatin (CDDP) and doxorubicin (DXR), used as positive controls, and evaluated by using the sulforhodamine-B (SRB) assay [20]. Briefly, cells were seeded into 96-well microtiter plates in 100 µL of the appropriate culture medium at plating densities at 2500, 5000 and 8000 cells/well for MCF-7, HCT-116 and SKOV-3, respectively, depending upon the doubling time of individual cell lines. After seeding, microtiter plates were incubated at 37 °C for 24 h before adding the test compounds. After 24 h, several samples of each cell line were fixed in situ with cold trichloroacetic acid (TCA) to represent a measurement of the cell population at the time of compound addition. The test compounds were freshly dissolved in dimethyl sulfoxide (DMSO, 10^−2^ M) and gradually diluted to different concentrations (0.79–50 µM) with a complete medium, so that the maximum DMSO/well ratio was 0.5% *v*/*v*. After the addition of different compound concentrations to triplicate wells, the plates were further incubated at 37 °C for 72 h. Cells were fixed in situ by the gentle addition of 50 µL of cold 50% *w*/*v* TCA (final concentration 10%) and incubated for 1 h at 4 °C. The supernatant was discarded, and the plates were washed with tap water and air-dried. SRB solution (100 µL) at 0.4% (*w*/*v*) in 1% acetic acid was added to each well, and the plates were incubated for 30 min at room temperature. After staining, the unbound dye was removed by washing with 1% acetic acid and the plates were air-dried. The bound stain was then solubilized with 10mM Trizma base and the absorbance was read on an automatic plate reader at 570 nm. The compound concentration able to inhibit cell growth by 50% (IC_50_ ± SD) was then calculated from semi-logarithmic dose–response plots.

### 3.3. Accession Codes

CCDCs 2225696, 2224240, 2224256 contain the supplementary crystallographic data for this paper. These data can be obtained free of charge via https://www.ccdc.cam.ac.uk/structures/ (accessed on 13 April 2023), or by emailing data_request@ccdc.cam.ac.uk, or by contacting The Cambridge Crystallographic Data Centre, 12 Union Road, Cambridge CB2 1EZ, U.K.; Fax: +44-1223-336033.

## 4. Conclusions

The study of the reactivity of 2,3-dihydro-2-methyl-1*H*-chromeno[3,2-*c*]pyridines (2,3-DHPCs, **2**) and 1,2,3,4-tetrahydro-2-methylchromeno[3,2-*c*]pyridin-10-one (1,2,3,4-THCP-10-ones, **1**), gave us the opportunity to synthesize novel compounds (**3**, **6** and the diastereomeric hemiacetals **7** and **8**) targeted at MAOs and ChEs, whose abnormal activity is implicated in neurological disorders, such as PD and AD. Regarding AD, although advances in understanding the multifactorial feature of the disease exist [21,22], its etiopathology still remains not completely clear, and to date, the available therapy is only symptomatic and essentially based on AChE (or BChE) inhibitors, with some promise coming from inhibitors of MAO B and amyloid beta (Aβ) plaque formation and aggregation. Over the last few decades, some of us, using suitable in silico (Q)SAR approaches, actively contributed to discovering novel inhibitors of MAO B [23,24], AChE/BChE and Aβ aggregation and toxicity [25,26]. In this context, the present study significantly added to our knowledge of potentially neuroprotectant small molecules.

Beyond some important SARs deduced from the biological evaluation of the newly synthesized compounds, all sharing a common 1*H*-chromeno[3,2-*c*]pyridine scaffold, potent MAO B inhibitors, namely **3a** and **3b**, were disclosed. Compound **3a**, also endowed with moderate activity against AChE/BChE, is a hit deserving further pharmacological studies, as a possible remedy in early symptoms of PD [27] and/or as an anti-oxidative neuroprotectant in AD patients [28].

While the neuroprotective effects of MAO inhibitors in neurological diseases have long been studied, albeit with a low success rate in terms of clinical entries, the role of MAOs in tumor insurgence and progression has been only recently reported [29]. The inhibition of tumor cell growth, as assessed for several newly synthesized derivatives in antiproliferative assays with MCF-7, HCT116 and SK-OV-3 cell lines, suggest that the 10-(1*H*-indol-3-yl)-bearing 2,3,4,10-THCP analog **6c** is noteworthy. Although these findings do not allow us to establish how much the inhibition of MAOs affects the antitumor effect, the combination of MAO inhibition with cytotoxicity toward tumor cells, such as that observed herein in **6a**–**c**, might represent an approach worthy of study in cancer treatment.

## Data Availability

Not applicable.

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
