# Peer review of "Chemical and Biological Evaluation of Novel 1H-Chromeno[3,2-c]pyridine Derivatives as MAO Inhibitors Endowed with Potential Anticancer Activity"

_ijms, 2023, doi:10.3390/ijms24097724_

Round 1

Reviewer 1 Report

The manuscript of Altomare and coworkers reports a study related to heterocycle compounds able to inhibit MAO and potentially exert anticancer activity.

The work is sound because it includes the chemical synthesis with details about the mechanisms of formation of hemiacetals, the crystallographic characterization useful also for the stereogenic assignement, the physico-chemical properties about the diasteromers, the biological evaluation with a inhibition insights on MAO-A, MAO-B, AChE and BuChE human targets, traditionally involved in neurological diseases. The proliferation effects were also studied on three cell lines.

Despite the good results, and considering the aim of the scientific project, that is related to combat against cancer in also by means of the multi-targeting approach and different methodologies, there are still minor issues that could be at least faced citing some missing references.

1.    Drug delivery systems

The formulation is an issue that could dramatically improve the effects of bioactive heterocyclic compounds such as those reported by the authors. A suggested example of a successful strategy was reported by Celano, M. et al Cytotoxic effects of a novel pyrazolopyrimidine derivative entrapped in liposomes in anaplastic thyroid cancer cells in vitro and in xenograft tumors in vivo Endocrine-Related Cancer, 2008, 15(2), pp. 499–510. The author should cite it in the introduction or the conclusion of the manuscript.

2.    Docking calculations

The compound presented by the authors exert their activities recognizing the enzymes MAO-A, MAO-B, AChE and BuChE as reported in table 3 and 4. Since for all those targets PDB models are available or buildable a docking study is appropriate to elucidate their mechanism of action as well as the rational design of novel derivatives. With this respect, no matter will be the efforts of the authors to perform docking experiments, it could be appropriate to cite at least one paper. Since the MAO inhibition is more pronounced for the compounds of this manuscript the authors could consider this reference Gidaro, M.C. et al Kaempferol as Selective Human MAO-A Inhibitor: Analytical Detection in Calabrian Red Wines, Biological and Molecular Modeling Studies Journal of Agricultural and Food Chemistry, 2016, 64(6), pp. 1394–1400. It could be inserted in the conclusions as further studies to be carried out on these compounds.

3.    Multi-targeting

The multi-targeting issue of the presented compounds is clear due their biological properties able to cover different applications. Moreover, other chromone-based derivatives have recently reported in literature by Reis, J. et al Multi-target-directed ligands for Alzheimer's disease: Discovery of chromone-based monoamine oxidase/cholinesterase inhibitors European Journal of Medicinal Chemistry, 2018, 158, pp. 781–800. So the authors might cite it maybe in the introduction.

In conclusion the manuscript needs to be updated with the above suggested issues and citations before recommending its publication on this journal.

Reviewer 2 Report

We consider the manuscript pertinent to the readers of this Journal’s Special Issue of “Enzymes and Enzyme Inhibitors—Applications in Medicine and Diagnosis”; Section of Molecular Pharmacology.

The study is the continuation of the previous work of the team, relating the synthesis of 1H-Chromeno[3,2-c] pyridine-derived compounds, aiming the inhibition of monoamine oxidase (MAO-A + B) and cholinesterases (AChE and BChE) enzymes. A clarification of the new compounds’ potential “Target-to-hit” for drug development was attempted.

The introduction covers both old and new references briefly and perfectly integrates the theme's main aspects.

This article is well written, with a good organization of the contents, only minor comments will be made concerning the content of conclusions.

Regarding the discussion of the results, we found it suitable. A very nice graphic pics/graphs quality accompanying the discussion increases the understanding of the discussed theme and clarifies the reasoning. We congratulate the authors on that!

The core experimental design of the chemistry part of the manuscript seems carefully elaborated and meticulously presented. However, regarding the methodology of biologic assays, some concerns were raised.

The description of the enzyme kinetic studies is insufficient to guarantee reproducibility. Despite being said based on previous protocols, it is necessary to make a brief description of some fundamental enzymological points, such as substrate concentrations, inhibitor concentrations, and enzyme concentrations. It should be clarified also how the enzyme activity was defined in the assay. In the cited references as the base protocol [ref.16,17_Line556], the inhibition kinetic was performed by linearization of the Michaelis Menten equation (L-B), thus resorting to the initial velocities. This manuscript [ijms-2339348] doesn´t present any type of inhibition constant estimated for discussion.

Furthermore, and still related to the inhibition methodology, and since the authors focus the entire study on the predictive performance as “Target-to-hit” for drug development, the inhibition studies based only on the IC50 parameter, raised profound concerns regarding the accuracy and precision of the findings.  The real potency of an inhibitor must be supported by inhibition constants (preferably accessing the real mechanism toward each enzyme) estimated by appropriate statistical tools.

So, the reproducibility of the results that support the conclusions as “Target-to-hit” might not be assured!!.

To support the reasoning behind our concerns a brief discussion is made below.

To optimize the drug for potency and specificity, researchers ought to focus on the appropriate kinetic studies on enzyme mechanism, and the value IC50 cannot provide that, once it is completely deprived of the information about the enzyme mechanism type.  

Literature has shown that 90% of drugs fail in clinical trials and it was noticed that the success rates for new development projects in Phase II trials have fallen from 28% to 18% (2008–2010) with insufficient efficacy being the most frequent reason for failure [Arrowsmith, J. Phase II failures: 2008–2010. Nat Rev Drug Discov 10, 328–329 (2011). https://doi.org/10.1038/nrd3439].

The reliability of published data on potential drug targets and reproducibility issues relating to this theme is discussed elsewhere [Scannell, J.W., Bosley, J., Hickman, J.A. et al. Predictive validity in drug discovery: what it is, why it matters and how to improve it. Nat Rev Drug Discov 21, 915–931 (2022). https://doi.org/10.1038/s41573-022-00552-x; Prinz, F., Schlange, T. & Asadullah, K. Believe it or not: how much can we rely on published data on potential drug targets? Nat Rev Drug Discov 10, 712 (2011). https://doi.org/10.1038/nrd3439-c1].

From what was exposed, it´s urgent to approach the potency of the drug with the most appropriate tools to access the inhibitor behavior towards the target enzyme, to minimize the failure of drug candidates in clinical trials.

The fact that the failures of drug candidates are extremely high, notwithstanding being considered potent inhibitors, indicates the limitations of the predictivity of the models. Moreover, the validity of those models (most of the time done assuming the IC50 values!!!) might be questionable.  The validity of those models and the reproducibility of the drug´s in vitro kinetic experiments are crucial issues to address if success rates in clinical trials are to be improved.

There may be several reasons for the observed lack of reproducibility. Among these, incorrect or inappropriate statistical analysis of results or insufficient sample sizes, which result in potentially high numbers of irreproducible or even false results, have been discussed [Ioannidis JPA (2005)Why Most Published Research Findings Are False. PLoS Med 2(8): e124. https://doi.org/10.1371/journal.pmed.0020124].

Specific Comment:

#__L587-632_ “5. Conclusions” _ most of the text presented in this section does seem to belong in the discussion. The conclusion item is not mandatory for this journal, however, a short concluding text is especially important when the discussion is complex, as seems to be the case. Dear authors, please provide a concise text of the main achievements in this study.

Final Comment:  dear authors, please include, in this study, real Michaelis Menten inhibition constants, using Non-Linear Regression, (not MM linearizations) to the detriment of IC50 values, which are absolutely devoid of any mechanistic foundation for the enzyme. A careful reading about the inappropriateness of IC50, especially for non-linear inhibitors (and mostly the more potent ones are non–linear!!!) can be done in the article below.

“There is renewed recognition that (mechanistic) enzymology is essential for establishing robust, reliable, and appropriate assays for screening campaigns to identify hit compounds that are also adaptable to comprehensively characterize inhibitors more fully than the traditional half-maximal inhibitory concentration (IC50) values alone. While useful, the information inherent in IC50 values does not indicate a change in inhibitory mechanism (for example, a shift from competitive to mixed-type inhibition) in a series of lead compounds and moreover, the data are poorly correlative with their pharmacodynamic action, as they give no measure of the period of time the compound ‘resides’ on its target”  in “Mechanistic enzymology in drug discovery: a fresh perspective” NATURE REVIEWS | DRUG DISCOVERY VOLUME 17 | FEBRUARY 2018 | 115_ [doi:10.1038/nrd.2017.219].

Round 2

Reviewer 2 Report

We appreciate the effort made by the authors in improving the methodology, concerning accuracy in the analysis of enzyme kinetics. Thank you!

The MM curves, despite being incomplete, show a situation compatible with Competitive Inhibition.

We asked the authors to clarify the text between lines 601-602, we thought maybe there was a typing error: "... the concentration producing 50% inhibition of the MAO activity (IC50) calculated by nonlinear regression." It is unclear whether the authors are referring to the IC50, which is not calculated by non-linear regression, or Kis estimation, in which the substrate affinity, Km [substrate concentration corresponding to 50% of the maximum reaction rate] is considered. Being the estimation of Kis we have suggested using non-linear regression.
